# Implementation of Triage System and Shortening Patient Journey Time to Prevent COVID-19 Transmission in a University Hospital during a Pandemic

**DOI:** 10.3390/ijerph18136996

**Published:** 2021-06-30

**Authors:** Chanon Kongkamol, Laaong Padungkul, Nuttanicha Rattanajarn, Supawich Srisara, Lalita Rangsinobpakhun, Kanarit Apiwan, Jittiwat Sompan, Chatchanok Prathipsawangwong, Pennapa Buathong, Sinat Chann, Pornchai Sathirapanya, Chutarat Sathirapanya

**Affiliations:** 1Department of Family and Preventive Medicine, Faculty of Medicine, Prince of Songkla University, Hat Yai, Songkhla 90110, Thailand; chanon.kon@psu.ac.th; 2Department of Nursing, Songklanagarind Hospital, Hat Yai, Songkhla 90110, Thailand; palaaong@medicine.psu.ac.th; 3Faculty of Medicine, Prince of Songkla University, Hat Yai, Songkhla 90110, Thailand; 5810310048@psu.ac.th (N.R.); 5910310153@psu.ac.th (S.S.); 5910310133@psu.ac.th (L.R.); 5910310016@psu.ac.th (K.A.); 5910310021@psu.ac.th (J.S.); 5910310034@psu.ac.th (C.P.); 5910310116@psu.ac.th (P.B.); 5910310184@psu.ac.th (S.C.); 4Department of Internal Medicine, Faculty of Medicine, Prince of Songkla University, Hat Yai, Songkhla 90110, Thailand; sporncha@medicine.psu.ac.th; 5Health Impact Assessment Research Center, Prince of Songkla University, Hat Yai, Songkhla 90110, Thailand

**Keywords:** COVID-19, PCR, patient under investigation, transmission, patient journey

## Abstract

To explore the characteristics of the patient under investigation (PUI), and the routes and the patient journey time in our outpatient service, we examined the demographic data, presenting symptoms, risks of contact with COVID-19 cases, and the results of real-time polymerase chain reaction (PCR) tests in PUI cases from March to May 2020. The contact time, transfer time and total journey time of patient journey routes in our hospital were also explored. The results were shown in numbers, percentages and medians (interquartile range, IQR). A total of 334 PUI cases were identified from our triage system. The median (IQR) age was 35 (27, 47) years. Cough was the most common presenting symptom (56.2%), while fever (≥37.5 °C) was found in only 19.8% of the cases. The median (IQR) time of onset of the presenting symptoms was 3 (1, 5) days. The most common risk of contact with COVID-19 cases found during the triage was living in or returning from an outbreak area. Fifteen (4.5%) of the PUI cases had positive real-time PCR tests. The contact time and transfer time were longest in the PUI ward and from the Emergency Department (ED) to the PUI ward, respectively. Plans and actions to shorten the transfer time between the ED and the PUI ward and the total journey time should be developed.

## 1. Introduction

Coronavirus disease 2019 (COVID-19), originally known as novel corona virus 2019 disease, is an acute respiratory infection with variable clinical severity. It was first reported in Wuhan City, China on 31 December 2019. COVID-19 was declared a global pandemic infection by the World Health Organization (WHO) on 11 March 2020 as the number of confirmed cases rapidly increased worldwide after the outbreak in China [1]. By the beginning of April 2020, the number of COVID-19 confirmed cases and the death toll rose dramatically, outnumbering those of both the SARS-CoV epidemic in 2003 and the MERS outbreak in 2012 [2]. The virus spreads mainly via respiratory droplets, for example, coughing and sneezing, especially when people stay in close contact with an infected person or gather in crowded or poorly ventilated places. The clinical symptoms of COVID-19 range from asymptomatic (30–40%) to loss of taste, and typical respiratory tract symptoms (57–67%) including fever, dry cough, dyspnea and acute respiratory distress syndrome (3%), which is potentially fatal [2]. The COVID-19 pandemic caused enormous adverse impact worldwide, not only a global health crisis with nearly 33 million confirmed cases and 1 million deaths in over 200 countries from January to September 2020, but also rapid economic recession [2]. Thailand also experienced a similar problem from a domestic outbreak after the surge of confirmed cases in the middle of March 2020 [3]. Starting from an entertainment pub in the Thonglor area and the Lumpinee Boxing Stadium in Bangkok as the primary epicenters of the infection, the number of COVID-19 confirmed cases grew rapidly, leading to the lockdown of public and business venues all over the country for disease transmission control, which was authorized by the Public Health Emergency Act. This affected middle- to low-income earners due to the widespread loss of regular earnings [3,4,5]. Moreover, the breakdown of the medical equipment production chain resulting from pandemic-associated economic disruption led to a shortage of medical devices and supplies necessary for the prevention of COVID-19 transmission and treatment. Meanwhile, like other parts of the country, the COVID-19 situation in southern Thailand became of greater concern. Besides the highest confirmed cases on 4 April 2020 in Phuket Province, a major tourist attraction in southern Thailand [6], an increased number of confirmed cases were reported in the three southern border provinces, i.e., Yala, Narathiwat, and Pattani. This was caused by the returning Islamic Dawah pilgrims from participation in a religious activity in Indonesia. Moreover, 60 confirmed cases from the Sadao District Immigration Center of Songkhla Province were added to the pool of COVID-19 cases from 25 April to 8 May 2020 [7,8].

A study from China during the COVID-19 outbreak showed that a well-planned and effective patient screening system in a medical center was necessary to maintain ongoing health care services effectively, and particularly to prevent COVID-19 outbreak within a hospital [9]. A study from South Korea reported the suspension of emergency department (ED) services after an escalating number of COVID-19 confirmed cases in the ED. After implementing newly revised COVID-19 patient triage criteria using patient characteristics and their clinical presentations, the situation dramatically improved [10]. Based on the results of the mentioned studies, the use of patient clinical characteristics to develop appropriate triage criteria for screening the attending patients of a medical center is useful. The limited availability of highly sensitivity and confirmatory COVID-19 tests such as real-time PCR at the beginning of the outbreak underlined the need for simple but highly sensitive screening criteria for the identification of suspected COVID-19 contact cases. Furthermore, a carefully planned and effective health care policy during a rapidly growing outbreak is necessary for the prevention of in-hospital transmission. At the beginning of the COVID-19 outbreak in March 2020, the screening criteria for suspected COVID-19 contact cases or patients under investigation (PUI) issued by the Ministry of Public Health, Thailand were widely implemented in hospitals around Thailand [11,12], as at that time, the real-time PCR test for the confirmation of a COVID-19 infection was strictly preserved for highly suspected COVID-19 cases only.

Songklanagarind Hospital is a university hospital at Prince of Songkla University, located in Songkhla Province in southern Thailand. The hospital is a tertiary medical center treating highly complicated cases in the area. Here, a hospital administration team including doctors, nurses, medical students and all allied health care providers launched a guideline for the prevention of COVID-19 transmission in the hospital at the start of the national outbreak. This study aimed to describe (a) the characteristics of the PUIs whose nasopharyngeal swab specimens were tested by real-time PCR for COVID-19, and (b) the outpatient triage system and the patient journey time following the carefully delineated routes while receiving hospital health services. We expect that the information of this study will be useful in developing more effective screening criteria and a triage system for more effective prevention of healthcare-associated transmission of COVID-19, either from patients to patients, or patients to health care workers (HCWs) and vice versa.

## 2. Materials and Methods

### 2.1. Study Setting and Participants

The study was performed in Songklanagarind Hospital, which is a 900-bed referral center in southern Thailand. All the outpatients who were triaged as “PUIs” and had real-time PCR tests for COVID-19 performed on their nasopharyngeal swab specimens between 1 March and 31 May 2020 were enrolled. We collected and analyzed the data from the hospital computerized medical records including patient demographic data, presenting characteristics, and underlying diseases such as hypertension, diabetes mellitus, chronic obstructive pulmonary disease (COPD), etc., which were diagnosed and treated in our or other hospitals. In addition, the routes of the outpatient journey during their visits at the Outpatient Department (OPD) or the Emergency Department (ED), time spent in each contact point (contact time), in transferring between contact points (transfer time) and total journey time were analyzed.

### 2.2. Routes of Patient Journey and Triage System in the Hospital

The routes of patient journey in the hospital during receiving regular services were adjusted for prevention of COVID-19 spreading in the hospital. Every route from each hospital entry point to the targeted contact points was carefully designed after repeated discussions between the hospital HCWs, the hospital administration board, and the Songklanagarind Hospital Committee for Controlling COVID-19 Infection (SCCCI). The principal aims of defining the specific routes for each targeted destination of health service were to minimize the number of contact points and to shorten the contact time as well as the transfer time. For this reason, the times of starting and ending the contact at each contact point from hospital entry to discharge from the hospital service were recorded. The details of the route of each patient’s journey, such as the number of contact points involved, the contact time at each contact point, and transfer time between each contact point, were described.

Every patient who attended Songklanagarind Hospital for medical services was screened for the risk of contact with COVID-19 cases at the screening points located at each hospital entrance before being allowed to enter the main hospital complex. Patients whose medical history met one of the screening criteria were triaged as PUIs, isolated from the other patients and promptly transferred to the screening center for respiratory infectious diseases (SCRID) for undergoing nasopharyngeal swabs and subsequent real-time PCR tests for COVID-19 infection. While waiting for the real-time PCR results, the PUI cases were admitted to a PUI ward. For avoiding any delay at a contact point, a well-trained transporter followed the carefully defined shortest route for transferring the PUI cases to the PUI ward. After the real-time PCR test results were reported through our hospital automated reporting system, depending on the test results, they would either be admitted to a COVID ward (positive result) or discharged (negative result) and requested to perform a 14-day self-quarantine at their homes (Figure 1). In COVID-19 confirmed cases, the treatment of COVID-19 was under the collaboration team, which was composed of hospital infectious disease, respiratory medicine, and critical care medicine physicians.

Patients who presented with acute respiratory tract symptoms but did not meet the criteria for PUI were transferred to the Acute Respiratory Infection (ARI) clinic for initial and short-term essential treatment and discharged from our service. They were encouraged to revisit our service only after the respiratory tract symptoms subsided. If risks of contact with the virus were finally determined after a repeated screening at the ARI clinic, the patients would be reclassified as PUIs and sent to SCRID for undergoing nasopharyngeal swabs (Figure 1).

### 2.3. Criteria for Patients under Investigation (PUIs)

The PUI criteria used in our hospital were adapted from those issued by The Ministry of Public Health, Thailand. Regular up-to-date modifications of the screening criteria from the consensus of serial meetings of the SCCCI were issued to keep up with the real situation of the outbreak at both regional and national levels. The PUI criteria announced by the SCCCI used during this study time were [11,12]:A history of fever or body temperature ≥37.5 °C at the hospital screening point;At least one upper respiratory tract symptom (e.g., cough, sore throat, runny nose, tachypnea, dyspnea) within 14 days prior to visiting the hospital;Any of the following risks of contact with COVID-19 cases within 14 days prior to the onset of the upper respiratory tract symptoms:
3.1.Recently traveled to or returned from abroad.3.2.Recently traveled to, returned from, or live in a current outbreak area of COVID-19.3.3.Living with an individual who recently returned from a current outbreak area of COVID-19.3.4.Practicing a high-risk occupation, i.e., contact with foreign tourists, or working in a crowded place.3.5.Recently visited one or more public places with a large gathering, e.g., local market, shopping mall, public transport system, religious practices venue, or any places announced by Provincial Communicable Disease Control Committee as a COVID-19 virus spreading epicenter.3.6.A history of close contact with a confirmed case of COVID-19.3.7.Working in a COVID ward, PUI ward, ED, screening point, SCRID, laboratory room, radiology unit, or as an anesthetist or a forensic physician.A patient who has pneumonia with a clinical suspicion of COVID-19.Being a member of a patient cluster with respiratory infection symptoms.

A patient who met any one of the following categories would be triaged as a PUI case in our screening system: (a) items 1 and 3, (b) items 2 and 3, (c) item 4, (d) item 5.

### 2.4. Specimen Collection for the Real-Time PCR Test

Like other hospitals in Thailand, our hospital used a real-time PCR test for COVID-19 as the confirmatory diagnostic test. The PUI cases were transferred to a negative pressure room in the SCRID, ED or COVID ward based on their clinical conditions to have nasopharyngeal swabs taken for real-time PCR tests. The hospital staff from respiratory medicine, infectious disease, critical care medicine, or the residency and fellowship trainees were assigned to collect the nasopharyngeal specimens after an intensive training course and under supervision of a senior staff physician or trainee. With the strict precautions against contracting and spreading the COVID-19 virus, negative pressure rooms were located near each screening point where the PUI cases were identified. During the specimen collection process for real-time PCR tests, all physicians were required to use personal protective equipment for prevention from COVID-19 contagion including wearing a N-95 face mask, face shield, long-cover and water-proof gown with hand gloves and hair-cover medical cap. The collected specimens from the PUI cases were immediately sent for real-time PCR tests in well-sealed containers. Whatever the first result was, a second confirmatory test would be performed. The total time spent completing the dual real-time PCR tests was 12–18 h initially, but it was later shortened to 8 h [13].

### 2.5. Health Care Worker Protection, Disinfection and Environmental Sanitation

A practical protocol for the hospital HCWs’ protection to prevent viral transmission in the hospital was put in place by the SCCCI. Apart from general self-hygiene care, such as social distancing, hand washing with alcohol-based gel provided everywhere within the hospital, and regular body temperature checks, wearing a medical-tape sealed face mask and a face shield while providing medical care in the OPD or ED was regulated and monitored by the head commander of every health care team. Additionally, regular disinfection and environmental sanitation according to the recommended disinfection protocols were applied in all hospital sectors. To reduce the possibility of contact with COVID-19 cases from another area and transmission in the hospital after returning to work, traveling out of the province needed to be approved by the hospital director. A daily self-report of personal health through a hospital web link was also obligatory.

### 2.6. Data Sources and Collection

We retrospectively collected the PUI demographic data, clinical symptoms of suspicion of COVID-19 infection, history of risks of contact with the virus and results of the initial investigations. All the data were retrieved from the computerized Hospital Information System (HIS) and the “novelcorona 2” data collection forms. Detailed data regarding the service and screening process from each unit such as the ARI clinic, SCRID and OPD were obtained from interviews with the HCWs on the sites, i.e., OPD or ARI clinic nurses or hospital infection control unit staff, etc. The collected data were validated, stored in the KOBO and kept in a computerized database for the final analysis.

### 2.7. Statistical Analysis

Descriptive statistics were used in this study. Categorical data were described as frequencies and percentages. Continuous data such as age, duration of symptoms, body temperatures, times from onset to specimen collections, contact time at each contact point, and transfer times during changing contact points were presented as medians and interquartile ranges (IQR). All statistical analyses were performed by R studio version 3.6.3.

### 2.8. Data Availability Statement

All data and statistical analysis methods used in this study are described in this article. There were no data deposited in other depository sources.

### 2.9. Ethical Considerations

Ethical approval of this study was granted by the Human Research Ethics Committee (HREC) of the Faculty of Medicine, Prince of Songkla University (REC.63-283-9-2). We strictly followed the regulations described in the 1964 Declaration of Helsinki and The International Conference on Harmonization in Good Clinical Practice. The accessibility of the patients’ medical information was limited to only our research team. All personal identification was concealed and deleted within 7 days according to the hospital’s personal information security policy. All analyses were carried out from fully anonymized and aggregated data to ensure patient confidentiality.

## 3. Results

### 3.1. Patient Demographic Data

A total of 814 patients were classified as PUIs and received COVID-19 real-time PCR tests from 1 March to 31 May 2020. We excluded 442 inpatients who had been planned for admission for elective therapeutic or diagnostic interventions, plus 38 cases with incomplete data records. Hence, 334 cases who were outpatients and visited our hospital for regular medical service were left for final analysis in this study.

Of all 334 patients, 190 were female (56.9%). The median (IQR) age of the study population was 35 years (IQR, 27–47). A total of 52.7% of the included cases practiced occupations with low COVID-19 contact risk, such as jobs that did not involve contact with foreign tourists or working in crowded places. HCWs were the largest proportion of the PUI cases who had occupations with high COVID-19 contact risk (23.6%). Hypertension was the most common underlying illness (11.13%) reported (Table 1).

### 3.2. Presenting Clinical Characteristics

The median time of presenting symptoms that were suspicious of COVID-19 was 3 (IQR, 1–5) days before visiting our hospital service. Fever (body temperature ≥ 37.5 °C) presented only in 66 of the 334 (19.8%) cases. Notably, 56.2% of the cases presented with cough, and 51.6% with sore throat. Five cases required endotracheal intubation (1.6%) due to severe respiratory distress. A history of living in or recently returning from a known COVID-19 outbreak area was the most frequently reported risk from the screening questions (48.3%) (Table 2).

### 3.3. Real-Time PCR Test and Imaging Results

The median time between the symptom onset to the nasopharyngeal and throat specimen collection was 3 (IQR, 2–7) days. Of all the real-time PCR tests for COVID-19, 15 of 334 tests (4.5%) were positive for COVID-19 virus, of which no hospital HCW was positive. Only 72 cases who had lower respiratory tract symptoms underwent chest radiography, which showed no significant abnormality in all cases (Table 2).

### 3.4. Routes of Patient Journey

There were 318 study cases whose journey routes in our hospital during this study time were available for review. There were four main hospital entry points and nine journey routes depending on the type of medical services required and the level of suspicion of contact with COVID-19 virus. Overall, it took approximately 30–60 min to complete a hospital visit, except when the admission to the COVID ward was required, which would take a longer time because of the waiting time for the real-time PCR test results.

### 3.5. Triage System and Special Contact Points for Prevention of COVID-19 Transmission

The number of regular hospital entrances was limited for screening and triage cases with risks of COVID-19 contact. In addition, redirecting the patients to follow the delineated routes was operated to reduce patient crowding in a specific area of our hospital. Screening questions and body temperature checks were performed at each hospital entry point to classify the visiting outpatients as (a) without respiratory symptoms and without COVID-19 contact risk, (b) with respiratory symptoms, or (c) with a high suspicion of COVID-19 contact (i.e., PUI) regardless of symptoms. After the screening, the patients were distributed to their next contact points according to their classifications.

The asymptomatic and without COVID-19 contact risk patients were allowed to receive regular hospital services, while those with symptoms of respiratory tract infection were transferred to the ARI clinic where the screening for the risks of COVID-19 contact was repeated. If there was a risk noted, the patients would be transferred to the SCRID immediately for nasopharyngeal swabs and subsequent real-time PCR tests. For cases that presented at our hospital during non-office hours, collection of the nasopharyngeal specimens would be performed in a negative pressure room at the ED. If a chest film was required during an evaluation, a portable radiography machine was used.

#### 3.5.1. Acute Respiratory Infection (ARI) Clinic

Patients who reported respiratory symptoms but did not meet the PUI criteria and had no emergency conditions (ESI: Emergency Severity Index ≥ 3) would be redirected from the regular hospital service route to the ARI clinic. After repeated triage, 119 cases of the ARI clinic were reclassified as PUIs, of which 103 (88.8%) and 16 (11.2%) cases were transferred to the SCRID and the ED for nasopharyngeal swabs, respectively.

The operation of the ARI clinic was supervised by an infectious control (IC) nurse who gave the final approvals for transferring the PUI cases from the ARI clinic to have nasopharyngeal swabs done in the SCRID or the ED. To utilize the limited medical resources efficiently and secure our reserved resources during the outbreak, an infectious disease physician was authorized to make a final decision in cases of controversial decision concerning PUI cases.

#### 3.5.2. Screening Center for Respiratory Infectious Diseases (SCRID)

The SCRID has regularly served as a screening center for community acquired respiratory infections. During the outbreak of COVID-19 in our region, it was used as a place for taking nasopharyngeal swabs for PUI cases. Ten of 103 (9.7%) PUI cases from the ARI clinic were admitted due to high suspicion of COVID-19 infection and for infection control. In contrast, no case from the OPD was admitted.

The time spent at each contact point (contact time) along with the time spent while transferring between contact points (transfer time) and the total time spent on each journey route (total journey time) are shown in Table 3.

By the implementation of the measures, there was no hospital outbreak of COVID-19 in our hospital as well as another tertiary hospital located nearby ours, in which a similar measure was applied. However, the details of patient journey and journey time were not studied in that center for comparison with the present study.

## 4. Discussion

The first outbreak of COVID-19 in Thailand started in the middle of January 2020, around one month after the outbreak in Wuhan, China. An immediate response plan to the outbreak, at both national and provincial levels, was implemented to counteract the rapid rise in the number of COVID-19 cases. Songkhla Province, which is bordered by Malaysia in the south, also experienced a high risk from the COVID-19 pandemic because of the immigrants from this neighboring country where the number of confirmed cases continuously increased. Illegal crossings of the border by the local people between the two countries markedly raised the risk of COVID-19 transmission across the border. Therefore, Songklanagarind Hospital was mandated to set up effective measures for the prevention of COVID-19 transmission in the hospital.

As in previous reports of COVID-19 confirmed cases, the median time of onset of the presenting symptoms was 3 (IQR, 1–5) days. Interestingly, upper respiratory tract symptoms such as cough (56.2%) and sore throat (51.6%) presented mostly, while fever was less (19.8%) in our study. Therefore, we suggest that screening for the cases with a risk of contracting COVID-19 by high body temperature was less sensitive. Additionally, lower respiratory tract symptoms such as dyspnea were not as common as reported. A study from a university hospital in Bangkok including 405 PUIs during the same COVID-19 pandemic as in our study reported that most of the cases were female (61.2%) and diabetes mellitus was the most common comorbidity (5.7%). Cough (72.5%), both dry and productive cough, fever (46.2%) and sore throat (45.7%) were the three most common presenting symptoms of the PUIs. Although anosmia was reported in the study in Bangkok (1.7%), our study found no cases of anosmia, but nasal congestion was reported in 38.8% of the cases. A higher percentage of headache (30.8% vs. 18.0%) was also noted in our study. Diarrhea was comparable between the two studies (10.9 vs. 11.5%) [14]. Moreover, among 4,404 PUIs admitted in a hospital in New Yok, the common presenting symptoms were cough (72%), fever (63%), dyspnea (43%), myalgia (23%), fatigue (14%) and diarrhea (14%) [15]. It is noteworthy that cough has been the most common presenting symptom of PUIs in many studies including the present study, but typical respiratory distress symptoms are more prominent in the study from the US than reported in the present study and the one from Bangkok. The requirement of intubation and intensive care unit admission was 3% and the overall death rate through the study period was 2.3% in the study from the US. In the present study, intubation and ICU admission were initiated in 1.6% of patients and no patient died through the study time, likely due to the milder respiratory symptoms of the PUIs in our study. For this reason, other social risks of contact with people with COVID-19 need to be considered concomitantly to triage a patient as a PUI case or not earlier, which will facilitate the control of disease transmission.

The majority of the patients visited our hospital in late March to early April (weeks 12–14 of the pandemic), which was the same time as the peak of the national pandemic from the boxing stadium cluster [16,17] (Figure 2). Additionally, the provincial outbreaks originating in the Jiranakhorn stadium [18] and Sadao immigration center [19] caused a rapid increase in the number of PUI patients. Our screening questions for COVID-19 infection showed that nearly half of the cases defined as PUI cases were patients who had a history of living in or returning from a COVID-19 outbreak area, followed by HCWs and those who had close contact with a person who had influenza-like symptoms or pneumonia. We suggest that the mentioned risks identified by the screening questions of possible contact with cases of COVID-19 are more sensitive than fever in detecting PUI cases. These criteria of sensitive screening are still in operation in the recent third wave of the COVID-19 pandemic in the year 2021 in Thailand.

Apart from the double screenings at the hospital entry points and outpatient service points that had been proved effective in some studies [9,10], we believe that shortening the patients’ exposure time in hospital can lessen the risk for COVID-19 outbreak in the hospital. The shortest contact time was in the ED because the rapid transfer of patients from the ED was necessary to reduce patient crowding in such a limited area for the prevention of viral transmission. Although many structural obstacles were removed, the green channel and simulation exercises for rapid patient transfer were repeatedly performed. After-action reviews of the health care process and patient transfer were also used to rearrange or modify the steps of each process for effective outcomes. The transfer time from the ED to the PUI ward (route 2, Table 3) was the longest because their locations are structurally distant in our center. Future management and building restructuring are advocated. Without the wait for an inpatient bed and basic blood test results requested at the ED, which were reported as major factors of delay in the ED in a study [20], the PUI cases in our hospital were promptly transferred after an essential medical evaluation to the PUI ward by a well-trained transporter, and the results of blood tests could be accessed through the hospital automated computerized reporting system. However, due to the necessity of having real-time PCR test results before discharge to perform home quarantine or admission to the COVID ward, the contact time in the PUI ward was inevitably the longest. We propose that these defined routes of patient journeys are useful for the prevention of COVID-19 transmission in our hospital. This principle is supported by the fact that no patient in the OPD had a positive real-time PCR test following our intensive screening and redirecting the patients with risks from the regular journey routes. Therefore, we suggest that well-delineated patient journey routes in accordance with their service needs should be implemented, while the standards and coverage of health care remain to prevent medical liability issues. A study reporting medical liability issues during the same pandemic time as ours in Italy showed a significant increase in medico-legal complaints of delayed treatment, hospitalization or hospital arrival and a lack of medical supervision of non-autonomous patients, particularly non-COVID-19 patients [21]. The HCWs’ fear of contracting COVID-19, a gap or uncertainty of available knowledge regarding viral infectivity and transmission, and shortages of personal protective equipment can cause delays or inappropriate health services that generated medico-legal prosecutions. Hence, medical liability protection or a “penal shield” for the HCWs working during the COVID-19 pandemic has been suggested in some studies [21,22,23]. We believe that scientifically reliable knowledge regarding COVID-19 infectivity and transmission provided for the hospital physicians and other HCWs is a useful strategy to eliminate unreasonable fear among them. In our hospital, our previous experiences in handling MERS-CoV and SARS-CoV cases taught us how to operate the triage system as well as personal protective measures for all HCWs. The hospital service policy during the pandemic insisted that all emergency cases and patients who really needed urgent support would be promptly treated based on the standards of medical care without delay. Therefore, no cases of medico-legal complaint occurred, and thus there was no need for medical liability protection for our HCWs at all.

For the HCWs’ safety and the security of all the workforce in our hospital during the pandemic, the SCCCI encouraged all hospital HCWs to strictly follow the guidelines of self-hygiene care and the practical protocols for the prevention of COVID-19 transmission issued by the committee. In Italy, rapidly rising numbers of confirmed cases, shortages of medical device supplies, particularly personal protective devices, and limited knowledge regarding individual protection during the early time of the pandemic resulted in high fatality of both the public and the HCWs, including physicians [24]. In fact, it is accepted that it is impossible for white coat warriors or heroes, which refer to all-level HCWs, to survive in this COVID-19 battle without barriers or armor. For this reason, knowledge regarding COVID-19 transmission was distributed and emphasized to all-level HCWs in our hospital by the committee before cooperation was requested or regulated if the health care procedures were at high risk of COVID-19 contact. By this way of practice, there were no cases of HCWs in our hospital reported with positive real-time PCR test for COVID-19.

We think that the PUI criteria should remain or be modified according to the current situation of each pandemic. Since many questions regarding the efficacy of the currently available COVID-19 vaccines, such as how long the sustainable efficacy of each COVID-19 vaccine is after complete vaccination, the efficacy against the emerging resistant viral strains and long-term safety, have not been answered clearly, besides the issues of equitable vaccine distribution and accessibility in Thailand, suitable measures for diligent prevention of COVID-19 transmission both in individuals and the health care service system should be strongly advocated. The implementation of a triage system and shortening the patient journey time while receiving medical services are useful for all levels of hospitals or medical centers for the prevention of contamination and transmission of COVID-19 among the HCWs and attending patients during a pandemic. The same principles of practice with some modifications based on the conditions faced at any one time or situation, if necessary, can facilitate to launch a timely standard/good practice guideline for the prevention of airborne transmission infections. Vaccination may not be the best way to deal with the recent or a future COVID-19 pandemic until a larger number of studies that confirm the efficacy and safety of the COVID-19 vaccines are available [25,26].

## 5. Conclusions

Effective and intensive screening by the history of recent COVID-19 contact risks, along with carefully predetermined patient journeys to lessen their exposure time in the hospital, are significant strategies for success in preventing intra-hospital transmission of COVID-19 or other future airborne infections.

## Figures and Tables

**Figure 1 ijerph-18-06996-f001:**
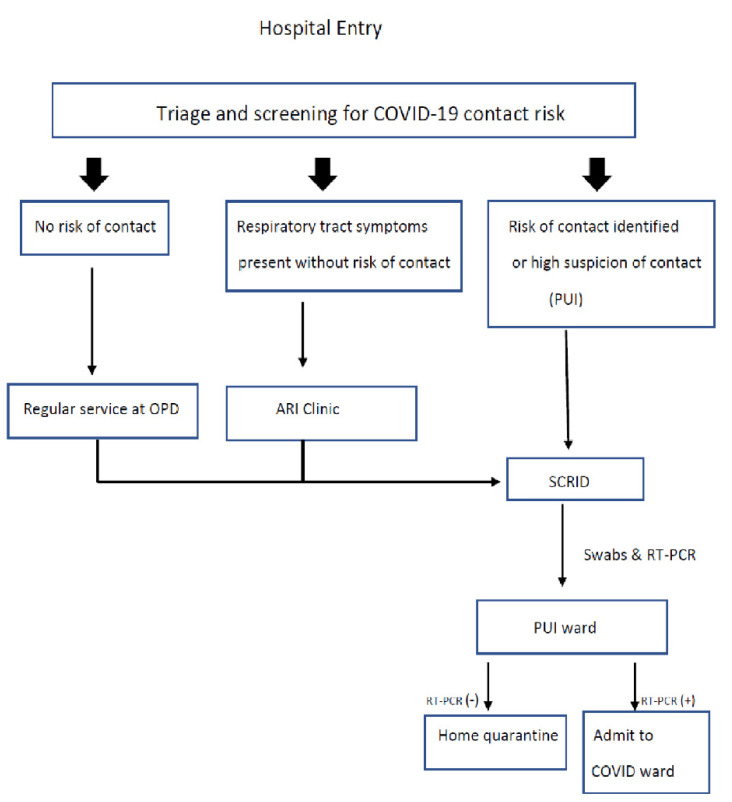
The hospital’s COVID-19 case screening and triage process. Abbreviations: OPD, Outpatient Department; ARI, Acute Respiratory Infection Clinic; SCRID, Screening Center for Respiratory Infectious Diseases; PUI, Patient Under Investigation; RT-PCR, real-time polymerase chain reaction.

**Figure 2 ijerph-18-06996-f002:**
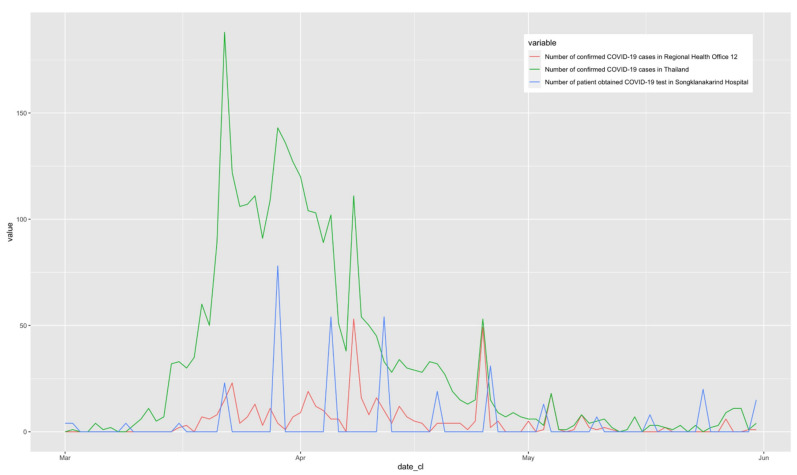
Number of the COVID-19 confirmed cases in the supervision area of Regional Health Office 12 in Thailand and number of patients who received real-time PCR tests for COVID-19 in Songklanakarind Hospital from 1 March to 31 May 2020. (The supervision area of Regional Health Office 12 covers the 7 lower provinces of southern Thailand including Songkhla, which is the current study site.)

**Table 1 ijerph-18-06996-t001:** Demographics and risk history of the patients under investigation (PUIs) and received real-time PCR tests for COVID-19 (n = 334, except indicated otherwise).

Gender	*n* (%)
Female	190 (56.9)
Male	144 (43.1)
**Age group**	
Age (years), median (IQR)	35 (27–47)
<12	17 (5.1)
12–18	5 (1.50)
19–35	152 (45.5)
36–60	125 (37.4)
>60	35 (10.5)
**Occupation: level of risk**	
1. Low-risk occupations	176 (52.7)
2. High-risk occupations	158 (47.3)
Health care worker	79 (23.6)
Work in an enclosed area (boxing stadium, night club, etc.)	2 (0.6)
Contact with foreigners/tourists	4 (1.2)
Multiple contacts with people (policemen, flight attendants, etc.)	44 (13.2)
Work in an area of COVID-19 contact risk (immigration officer, international air port, etc.)	20 (6.0)
Service worker (massager, barber, etc.)	9 (2.7)
**Residence**	
Songkhla	246 (73.7)
Yala	12 (3.6)
Pattani	12 (3.6)
Narathiwat	8 (2.4)
Phatthalung	13 (3.9)
Nakhon Si Thammarat	8 (2.4)
Suratthani	6 (1.8)
Satun	6 (1.8)
Trang	6 (1.8)
Phuket	3 (0.9)
Chumphon	1(0.3)
Bangkok	7 (2.1)
Other provinces	4 (1.2)
Foreign countries	2 (0.6)
**Religions**	
Buddhist	266 (79.6)
Islam	64 (19.2)
Other	4 (1.2)
**Payment sources**	
Universal Coverage	39 (11.7)
Civil Servant Medical Benefit Scheme	102 (30.5)
Government or State Enterprise	5 (1.5)
Social Security Scheme	68 (20.4)
Self payment	106 (31.7)
Other	14 (4.2)
**Underlying diseases** (≥1) (*n* = 256, *total number of cases* = 64)	
Allergic rhinitis	14 (5.5)
Chronic obstructive pulmonary disease	4 (1.6)
Asthma	6 (2.3)
Pulmonary tuberculosis	4 (1.6)
Diabetes	14 (5.5)
Hypertension	29 (11.3)
Coronary heart disease	7 (2.7)
Hepatitis	1 (0.4)
Chronic kidney disease	3 (1.2)
Any malignancies	13 (5.1)
**Risk history**	
Contact with poultry with risk of SARS-CoV2 infection	8 of 331 (2.4)
Contact with mammal with risk of SARS-CoV2 infection	56 of 331 (17.8)
History of visiting poultry/wild animal/mammal/seafood market	40 of 323 (12.4)
Living in or returning from an outbreak area	160 of 331 (48.3)
History of caring for or having close contact with one or more people who had influenza-like symptoms or pneumonia	71 of 330 (21.5)
History of severe pneumonia or unknown cause of death	10 of 331 (3.0)
History of getting treated at or contact with a health care worker or a patient admitted in a hospital of a pandemic area	33 of 331 (10.0)
Health care worker or laboratory staff	82 of 331 (24.8)
A member of a cluster of patients with pneumonia	9 of 331 (2.7)

**Table 2 ijerph-18-06996-t002:** Clinical characteristics, initial diagnosis, specimen collection, investigation and laboratory findings.

Clinical characteristics (*n* = 322 except for Fever *n* = 334)	*n* (%)
Duration of presenting symptom (days), median (IQR)	3 (1–5)
Body temperature (°C), median (IQR)	37.0 (36.7–37.4)
Fever (≥37.5 °C)	66 (19.8)
Cough (dry and productive)	181 (56.2)
Sore throat	166 (51.6)
Myalgia	100 (31.1)
Nasal discharge	125 (38.8)
Sputum production	120 (37.3)
Dyspnea	52 (16.2)
Headache	99 (30.8)
Diarrhea	35 (10.9)
Required intubation	5 (1.6)
**Initial diagnosis (*n* = 296)**	
Upper respiratory tract infection	217 (73.3)
Pneumonia	20 (6.8)
Other	59 (19.9)
**Specimen Collection (*n* = 334)**	
Duration from the symptom onset to specimen collection (days), median (IQR)	3 (2–7)
nasal swab	334 (100.0)
throat swab	334 (100.0)
Sputum	6 (1.8)
tracheal suction	3 (0.9)
Saliva	3 (0.9)
**Investigation and laboratory findings**	
Chest film, (*n* = 308)	72 (23.4)
Blood leukocyte count (×10^3^/μL), (*n* = 51)	
<4	3 (5.9)
>10	14 (27.5)
Neutrophil count (×10^3^/μL), (*n* = 43)	
<2	2 (4.7)
>6.5	13 (30.2)
Lymphocyte count (×10^3^/μL), (*n* = 43)	
>1.5	21 (48.8)
0.5–1.5	17 (39.5)
<0.5	5 (11.6)
Platelet count < 150 (×10^3^/μL), (*n* = 50)	3 (6)

**Table 3 ijerph-18-06996-t003:** Contact time, transfer time and total patient journey time in each route.

Route	*n*	Contact 1	Contact Time (min), Median (IQR)	Transfer Time (min), Median (IQR)	Contact 2	Contact Time (min), Median (IQR)	Transfer Time (min), Median (IQR)	Contact 3	Contact Time (min), Median (IQR)		Total Journey Time (min)
No.1	18	ED	16.50 (13.00–33.00)	Discharge							16.5
No.2	16	ED	16.50 (13.00–33.00)	161.00 (101.00–216.00)	Ward	333.00 (106.00–574.50)	Admission				510.5
No.3	15	OPD	53.00 (17.75–115.50)	16.00 (11.00–21.00)	SCRID	31.50 (19.00–47.00)	Discharge				100.5
No.4	2	OPD	53.00 (17.75–115.50)	12.00	ARI	19.00 (13.00–33.00)	2.00	SCRID	31.50 (19.00–47.00)	Discharge	117.5
No.5	134	SCRID	31.50 (19.00–47.00)	Discharge							31.5
No.6	14	SCRID	31.50 (19.00–47.00)	30.00 (7.00–63.00)	Ward	333.00 (106.00–574.50)	Admission				394
No.7	16	ARI	19.00 (13.00–33.00)	25.00 (11.50–39.50)	ED	16.50 (13.00–33.00)	Discharge				60.5
No.8	93	ARI	19.00 (13.00–33.00)	24.00 (16.00–34.00)	SCRID	31.50 (19.00–47.00)	Discharge				74.5
No.9	10	ARI	19.00 (13.00–33.00)	24.00 (16.00–34.00)	SCRID	31.50 (19.00–47.00)	15.50 (7.00–22.00)	Ward	333.00 (106.00–574.50)	Admission	423

Abbreviations: ED, Emergency Department; OPD, Outpatient Department; ARI, Acute Respiratory Infection Clinic; SCRID, Screening Center for Respiratory Infectious Diseases; Ward, the PUI Ward.

## Data Availability

All data and statistical analysis methods used in this study were included in this article. No data were deposited in other depository sources.

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
