# Peer review of "Implementation of Triage System and Shortening Patient Journey Time to Prevent COVID-19 Transmission in a University Hospital during a Pandemic"

_ijerph, 2021, doi:10.3390/ijerph18136996_

Round 1
Reviewer 1 Report
Kongkamol and collaborators aimed to investigate the demographic and clinical characteristics of patients with potential COVID-19 from March to May 2020 in a university hospital in Thailand. The research topic is important for improving the health attention of COVID-19 patients particularly in hospitals with limited resources. However, I consider this manuscript requires some modifications in order to make it more clear. Specifically, I have the following recommendations:
- The sentences from line 30 and line 36 of the Abstract section are repetitive. I recommend removing one of them
- I suggest including references for the sentences of Line 57, 59, and 62
- The abbreviation for People’s Republic of China (PRC) defined in line 47 does not seem to correspond to the same as in line 77 - PRC. I recommend authors to double check and modify these abbreviations to avoid using the same abbreviation for two different terms
- I suggest mentioning the previous studies performed in the description of patients with potential COVID-19. What is new in this study compared with previous studies?
- In line 136, the abbreviation for the Patients patient under investigation is defined. However, this abbreviation was already used and not defined in line 114
- It is not clear how comorbidities were evaluated. I suggest providing more details of the comorbidities evaluation
- I suggest discussing the obtained results with those from previous studies studying the same outcomes.
- Now that the COVID-19 vaccine is available, do you think the PUI criteria should be modified? If so, it would be worthy of mention in the Discussion
- It might be worthy to discuss how the findings of this study could be translated into specific measures to improve the health care of patients with potential COVID-19. Also, whether this translation could be applied to settings different to the one described in the study (a university hospital)
Author Response
Response to Reviewer 1 Comments and Suggestions for Authors
Kongkamol and collaborators aimed to investigate the demographic and clinical characteristics of patients with potential COVID-19 from March to May 2020 in a university hospital in Thailand. The research topic is important for improving the health attention of COVID-19 patients particularly in hospitals with limited resources. However, I consider this manuscript requires some modifications in order to make it more clear. Specifically, I have the following recommendations:
- The sentences from line 30 and line 36 of the Abstract section are repetitive. I recommend removing one of them
Response: the sentence in line 35 -36 was removed.
- I suggest including references for the sentences of Line 57, 59, and 62
Response: The corresponding references were added (line 55, 58 and 60).
- The abbreviation for People’s Republic of China (PRC) defined in line 47 does not seem to correspond to the same as in line 77 - PRC. I recommend authors to double check and modify these abbreviations to avoid using the same abbreviation for two different terms
Response: “China” replaced “People’s Republic of China (PRC)” in both places (line 45 and 76).
- I suggest mentioning the previous studies performed in the description of patients with potential COVID-19. What is new in this study compared with previous studies?
Response: We add the following content in discussion section
“A study from a university hospital in Bangkok including 405 PUIs during the same time of COVID-19 pandemic as that in our study showed that most the cases were female (61.2%) and diabetes mellitus was the most common comorbidity (5.7%). Cough (72.5%), both dry and productive cough, fever (46.2%) and sore throat (45.7%) were the three most common presenting symptoms of PUIs. Our study reported higher percentage of cough (93.5%) but lower percentage of fever (19.8%). Whereas anosmia was reported in the study in Bangkok (1.7%), our study found no case of anosmia, but nasal congestion was reported in 38.8%. Higher percentage of headache (30.8% vs.18.0%) was also noted was in our study. Diarrhea was comparable between the two studies (10.9 vs. 11.5%).” [Bruminhent J et al., Clinical characteristics and risk factors for corona virus disease (COVID-19) among patients under investigation in Thailand. PLOS ONE September 15, 2020] (line 371-379 in Discussion)
- In line 136, the abbreviation for the Patients patient under investigation is defined. However, this abbreviation was already used and not defined in line 114
Response: PUI was defined in line 113, and only the abbreviation “PUI” was used in line 136.
- It is not clear how comorbidities were evaluated. I suggest providing more details of the comorbidities evaluation
Response: The comorbidities (underlying diseases) in the current study were based on the information in the medical records of our hospital, or the treatments received for either comorbidity mentioned from the other centers. The standard criteria for each comorbidity was used for the diagnosis of each comorbidity (line 114-116, in Materials &Methods).
- I suggest discussing the obtained results with those from previous studies studying the same outcomes.
Response: as in the item 4
We also add this content from a study from a NY hospital describing the characteristics of PUIs in the western world, in addition to that we described in response to the reviewer’s comment/suggestion item no. 4
“Among 4,404 PUIs admitted in a hospital in New Yok, the most common presenting symptoms were cough (72%), fever (63%), dyspnea (43%), myalgia (23%), fatigue (14%) and diarrhea (14%). [Singer AJ., et al. Cohort of 4404 PUIs for COVID-19 in a NY hospital and predictors of ICU care and ventilation. Infectious dis 2020;76:394-404.] It is noteworthy that cough is the most common presenting symptoms of PUIs in many studies including the present study, but typical respiratory distress symptoms are more prominent in the study from the US than that reported in the present study, and the one in Bangkok. Requirement of intubation and intensive care unit admission is 3 % and the overall death rate through the study period is 2.3% in the study from the US. In the present study, intubation and ICU admission was initiated in 1.6%, and no case was dead through the study time. This would be due to milder respiratory symptoms of PUIs in our study.” (line 379-389 in Discussion)
- Now that the COVID-19 vaccine is available, do you think the PUI criteria should be modified? If so, it would be worthy of mention in the Discussion
Response: The authors think that the PUI criteria should be remained or modified according to the current situation of each pandemic. Since many questions such as how long the efficacy of each COVID-19 vaccine sustains after complete vaccination, the efficacy against the emerging resistant viral strains, and long-term safety have not been answered clearly besides the equitable vaccine distribution and accessibility, suitable measures for diligent prevention of COVID-19 transmission both in individual and health care service system should be strongly advocated. Vaccination may not be a well-reliable solution to end a current or future COVID-19 pandemic.
[Burgos RM, et al. The race to a COVID vaccine:……..2021;2020-12-2.]
[Babari A. COVID-19 vaccine concerns: Facts or Fictions. Experiment and Clin Transplant 2021.DOI:10.6002/ect.2020.0056] (This was mentioned in the last paragraph of Discussion, line 471-486)
- It might be worthy to discuss how the findings of this study could be translated into specific measures to improve the health care of patients with potential COVID-19. Also, whether this translation could be applied to settings different to the one described in the study (a university hospital)
Response: The authors think that implementation of triage system and shortening patient journey time during receiving medical services are useful for all levels of hospitals or medical centers to prevention of contamination and transmission of COVID-19 among the health care providers and the attending patients, during a pandemic of either the recent COVID-19 or other air-borne infections in the future. As the same principles of practice with some modification based on the facing conditions or situations, if necessary, can facilitate to launch a standard/good practice guideline timely for prevention of air-borne transmission diseases. (This was mentioned in the last paragraph of Discussion, line 478-484)
Reviewer 2 Report
the article by Kongkamol et al., is well developed and presented some doubts and suggestions:
- put the tables close to where they are described
- use another format for the tables
- What is the purpose of figure 2?
- In general, the article is fine, it is difficult to follow the ideas of the authors since the figures and figures are wrongly placed in the text.
Author Response
Response to Reviewer 2’s comment
Comments and Suggestions for Authors
the article by Kongkamol et al., is well developed and presented some doubts and suggestions:
- put the tables close to where they are described
Response: Done, as in the revised version submitted to the journal.
- use another format for the tables
Response: Done
- What is the purpose of figure 2?
Response:- Figure 2 was removed, because it has the same meaning as Table 3, to avoid the repetitive content
- In general, the article is fine, it is difficult to follow the ideas of the authors since the figures and figures are wrongly placed in the text.
Response: The Figures and Tables were re-placed to where they are referred to.
Reviewer 3 Report
I read with great interest the work entitled “Implementation of Triage System and Shortening Patient Journey Time to Prevent COVID-19 Transmission in a University Hospital During a Pandemic".
The work proposes organizational solutions to reduce intra-hospital respiratory transmission.
Overall the work is well written and the proposed protocol is adequate for the purpose.
I ask the authors some small but fundamental clarifications to make the paper publishable.
Major Concerns
- In addition to a comparison with national data, information on the progress of the pandemic in the specific area of ​​the study (southern Thailand or city where the hospital is located) would be needed.
- The study proposes a prevention strategy and amply illustrates its characteristics. However, a fundamental fact is missing from the work. The authors do not describe whether or not there have been hospital outbreaks in the hospital. From the contents it is assumed no but this data should be made explicit in the results.
- To evaluate the effectiveness of the method, a control sample would also be needed (e.g. other tertiary hospital where different strategies have been applied).
- To reinforce the importance of their study, the authors must also mention the effects of a fast and efficient triage also on other pathologies. In other situations, the shortage in this area has also created considerable problems of a medical-legal nature [they should mention for this purpose in the introduction or discussion Nioi M, et al. Fear of the COVID-19 and medical liability. Insights from a series of 130 consecutives medico-legal claims evaluated in a single institution during SARS-CoV-2-related pandemic. Signa Vitae. 2021. doi:10.22514/sv.2021.098d'Aloja, Ernesto, et al. "COVID-19 and medical liability: Italy denies the shield to its heroes." EClinicalMedicine 25 (2020).]
- The speeding up of contact times is also an indirect protection for Healthcare Workers. In the article you mention this aspect but a small extension of the discussion would be needed. [cfr Nioi, Matteo, et al. "COVID-19 and Italian healthcare workers from the initial sacrifice to the mRNA vaccine: Pandemic chrono-history, epidemiological data, ethical dilemmas, and future challenges." Frontiers in Public Health 8 (2020).]
The simple description of the strategy without the description of the results obtained and without a comparison with the data of a tertiary center that has adopted a different strategy is not sufficient to make the study suitable for publication.
Minor concerns.
The number of citations should be increased by making particular reference to articles that speak of centers adopting different strategies. A sub-chapter in the discussion could be dedicated to this topic.
Author Response
Response to Reviewer 3’s comments
Comments and Suggestions for Authors
I read with great interest the work entitled “Implementation of Triage System and Shortening Patient Journey Time to Prevent COVID-19 Transmission in a University Hospital During a Pandemic".
The work proposes organizational solutions to reduce intra-hospital respiratory transmission.
Overall the work is well written and the proposed protocol is adequate for the purpose.
I ask the authors some small but fundamental clarifications to make the paper publishable.
Major Concerns
- In addition to a comparison with national data, information on the progress of the pandemic in the specific area of ​​the study (southern Thailand or city where the hospital is located) would be needed.
Response : Done, as shown in Fig. 2
- The study proposes a prevention strategy and amply illustrates its characteristics. However, a fundamental fact is missing from the work. The authors do not describe whether or not there have been hospital outbreaks in the hospital. From the contents it is assumed no but this data should be made explicit in the results.
Response: Yes, there was no outbreak in our hospital. We addressed this point in the RESULT.
“ By implementation of these measures, there was no hospital outbreak of COVID-19 in our
hospital as well as another tertiary hospital located nearby ours, in which a similar measure was
applied. However, the patient journey time was not studied in that center for comparison with the
present study.” (line 351-354 -thelast paragraph of results before Discussion)
- To evaluate the effectiveness of the method, a control sample would also be needed (e.g. other tertiary hospital where different strategies have been applied).
Response: There is a tertiary hospital in this area where a similar strategic plan was applied but the data of patient journey time was recorded. No in-hospital COVID-19 outbreak was reported in the other tertiary hospital also. (as the response in Item 2 above)
- To reinforce the importance of their study, the authors must also mention the effects of a fast and efficient triage also on other pathologies. In other situations, the shortage in this area has also created considerable problems of a medical-legal nature [they should mention for this purpose in the introduction or discussion Nioi M, et al. Fear of the COVID-19 and medical liability. Insights from a series of 130 consecutives medico-legal claims evaluated in a single institution during SARS-CoV-2-related pandemic. Signa Vitae. 2021. doi:10.22514/sv.2021.098. d'Aloja, Ernesto, et al. "COVID-19 and medical liability: Italy denies the shield to its heroes." EClinicalMedicine 25 (2020).]
Response: We added this content in Discussion:
“A study reporting medical liability issues during the same time of pandemic with ours in Italy showed a significant increase in medico-legal complaints of delay treatment, hospitalization or hospital arrival and lack of medical supervision of non-autonomous patients, particularly, the non-COVID-19 patients. [Nioi M, et al. Fear of the COVID-19 and medical liability. Insights from a series of 130 consecutives medico-legal claims evaluated in a single institution during SARS-CoV-2-related pandemic. Signa Vitae. 2021. doi:10.22514/sv.2021.098.] The HCWs’ fear of contracting COVID-19, a gap or uncertainty of available knowledge of viral infectivity and transmission, and shortage personal prevention equipment can cause delay or inappropriate health service that generated medico-legal prosecutions. Hence, medical liability protection or “penal shield” for the HCWs working during a COVID-19 pandemic has been suggested. [Nioi M, et al. Fear of the COVID-19 and medical liability. Insights from a series of 130 consecutives medico-legal claims evaluated in a single institution during SARS-CoV-2-related pandemic. Signa Vitae. 2021. doi:10.22514/sv.2021.098.; d'Aloja, Ernesto, et al. "COVID-19 and medical liability: Italy denies the shield to its heroes." EClinicalMedicine 25 (2020); Cioffi A. Covid-19 and medical liability: A delicate balance. Medco-legal J 2020;88:187-8.] The authors think that scientifically reliable knowledge regarding COVID-19 infectivity and transmission provided for the hospital physicians and other HCWs is a crucial strategy to eliminate the unreasonable fear among them. In addition, our experience in handling MERS-CoV and SARS-CoV cases taught us how to operate the triage system as well as personal protection measures. The hospital service policy at the time of this study insisted that all emergency cases and patients who really need urgent support will be promptly treated based on the standard of medical care without delay.” (line 439-456, in Discussion)
- The speeding up of contact times is also an indirect protection for Healthcare Workers. In the article you mention this aspect but a small extension of the discussion would be needed. [cfr Nioi, Matteo, et al. "COVID-19 and Italian healthcare workers from the initial sacrifice to the mRNA vaccine: Pandemic chrono-history, epidemiological data, ethical dilemmas, and future challenges." Frontiers in Public Health 8 (2020).]
Response: Health care workers protection measures are mentioned like:
1.1 In section 2.4 of material and methods
“…… During the specimen collection process for RT-PCR test, all physicians were regulated to perform full self-protection from COVID-19 contagion such as wearing N-95 face mask, face shielding, long-cover and water-proof gown with hand gloves, and hair-cover medical cap…..” (line 201-204)
1.2 Section 2.5 was added as follow:
2.5. Health care provider protection, disinfection and environmental sanitation
A protocol of practice for the hospital health care workers to prevent the viral trans
mission in the hospital was launched by the SCCCI. Apart from general self-hygiene care,
such as social distancing, hand washing with alcohol-based gel provided everywhere
within the hospital, and regular body temperature checks, wearing medical-tape sealed
face mask and face shield during providing medical care in OPD or ED were regulated and
monitored by the head commander of every health care team. Also, regular disinfection and
environmental sanitation according to the recommended disinfection protocols were ap
plied in all hospital sectors. To reduce the possibility of contact COVID-19 from another
area and transmission in the hospital after returning to work, travelling out of the province
must be approved by the hospital director. A daily self-report of personal health through
hospital web link is obligatory.
- In Discussion
“ For the HCWs’ safety and security in our hospital, the SCCCI encourages the application of self-hygiene care and strict compliance to the guidelines for prevention of the COVID-19 transmission issued by the committee. Rapidly rising number of the confirmed cases, shortage of medical device supply, particularly, personal protection devices, and limitation of knowledge regarding individual protection during the early pandemic time resulted in high fatality both the public and the HCWs including physicians [Nioi M., et al. COVID-19 and Italian healthcare workers from the initial sacrifice to the mRNA vaccine: Pandemic chrono-history, epidemiological data, ethical dilemmas, and future challenges. Frontiers in Public Health 8 (2020)]. In fact, it is impossible for a white coat warrior to survive in the battle without barrier or armor. For this reason, knowledge regarding the COVID-19 transmission was distributed and emphasized to all-level HCWs and alliances in our hospital by the committee, before cooperation was requested or regulated if the actions were critical. By this way of practice, there was no case of HCW in our hospital reported with positive RT-PCR for COVID-19.” (line 457-470)
The simple description of the strategy without the description of the results obtained and without a comparison with the data of a tertiary center that has adopted a different strategy is not sufficient to make the study suitable for publication.
Response: as in item 2 and 3 above
Minor concerns.
The number of citations should be increased by making particular reference to articles that speak of centers adopting different strategies. A sub-chapter in the discussion could be dedicated to this topic.
Response: New references were added to make the concept and application of the plan discussed in this article more understandable.
Round 2
Reviewer 1 Report
The authors have modified their manuscript according to the reviewer's suggestions. I do not have more comments or suggestions about the text.
Reviewer 3 Report
I thank the authors for making the required changes. I think the paper has improved considerably now. I have no further comments to make.